# The Right to Ask, the Need to Answer—When Patients Meet Research: How to Cope with Time

**DOI:** 10.3390/ijerph20054573

**Published:** 2023-03-04

**Authors:** Manuela Priolo, Marco Tartaglia

**Affiliations:** 1Unità di Genetica Medica, Grande Ospedale Metropolitano Bianchi-Melacrino-Morelli, 89124 Reggio Calabria, Italy; 2Genetica Molecolare e Genomica Funzionale, Ospedale Pediatrico Bambino Gesù, IRCCS, 00146 Rome, Italy

**Keywords:** rare diseases, diagnosis, physician-patient relationship, diagnostic odyssey, diagnostic delay, orphan diseases, undiagnosed patients

## Abstract

Reaching a diagnosis and its communication are two of the most meaningful events in the physician–patient relationship. When facing a disease, most of the patients’ expectations rely on the hope that their clinicians would be able to understand the cause of their illness and eventually end it. Rare diseases are a peculiar subset of conditions in which the search for a diagnosis might reveal a long and painful journey scattered by doubts and requiring, in most cases, a long waiting time. For many individuals affected by a rare disease, turning to research might represent their last chance to obtain an answer to their questions. Time is the worst enemy, threatening to disrupt the fragile balance among affected individuals, their referring physicians, and researchers. It is consuming at all levels, draining economic, emotional, and social resources, and triggering unpredictable reactions in each stakeholder group. Managing waiting time is one of the most burdensome tasks for all the parties playing a role in the search for a diagnosis: the patients and their referring physicians urge to obtain a diagnosis in order to know the condition they are dealing with and establish proper management, respectively. On the other hand, researchers need to be objective and scientifically act to give a rigorous answer to their demands. While moving towards the same goal, patients, clinicians, and researchers might have different expectations and perceive the same waiting time as differently hard or tolerable. The lack of information on mutual needs and the absence of effective communication among the parties are the most common mechanisms of the failure of the therapeutic alliance that risk compromising the common goal of a proper diagnosis. In the landscape of modern medicine that goes faster and claims high standards of cure, rare diseases represent an exception where physicians and researchers should learn to cope with time in order to care for patients.

## 1. Introduction

The diagnosis of a disease still remains one of the most important moments in medicine and in patients’ lives. It explains the co-occurrence of many symptoms, directs the appropriate treatment, and gives the possibility of a prognosis. The communication of a diagnosis takes on an even more relevant role in the context of chronic disabling diseases as it makes it possible to define a condition that will accompany the patient for the rest of their life and to establish appropriate rehabilitation or symptomatic support when effective therapy is not possible. Rare diseases (RDs) represent a particular subgroup in chronic disabling diseases. Frequently genetic in origin and often childhood-onset, they carry a significant emotional burden that falls on patients, families, and, their referring physicians [1]. The current technological and scientific expertise in genomics, and more general research, has significantly impacted the diagnostic rate in RDs by improving the ability to identify pathogenic variants through the application of parallel sequencing. Moreover, extensive patient and physician empowerment has ensured increased and more equal access to diagnostic services [2]. Finally, international connection with data-sharing programs has multiplied the likelihood of information exchange and validation of the causative role of the identified variants through matchmaking platforms that are able to create repositories of undiagnosed cases [3,4]. Among the international initiatives, many conjoined programs have settled ambitious goals to be achieved in the next years for assuring a rapid diagnosis of RDs. The future of RD research has been well outlined by the International Rare Diseases Research Consortium (IRDiRC), which has proposed three major goals in order to achieve proper diagnosis and treatment in RDs and clearly states as Goal 1: “*All patients coming to medical attention with a suspected rare disease will be diagnosed within 1 year if their disorder is known in the medical literature; all currently undiagnosable individuals will enter a globally coordinated diagnostic and research pipeline*” [5]. Unfortunately, the reality is so far from this ambitious goal that, actually, achieving a diagnosis is a road riddled with dark sides and pitfalls. Alongside the consolidated literature about the economic burden and the impact on the quality of life of patients affected with a chronic illness [6,7,8,9], the ethical, legal, social, and psychological implications (ELSI and ELPAG) of RD are now a growing issue in medicine and genomic research, as evidenced by the widespread initiatives on ELSI and ELPAG research supported by human genetics scientific societies [10,11,12] and other international research programs, such as the ELSI Research Program at the National Human Genome Research Institute [13]. Numerous initiatives are available with strong educational components to provide genomic education to healthcare professionals and empower patients and associations. Online networks, (e.g., Dyscerne [14]), expert resources for rare conditions (e.g., Orphanet [15]), and the educational programs of the European Reference Networks (ERN) are now available for non-genetics specialties and led by geneticists [16,17,18]. However, very little is known about the psychological consequences and ethical dilemmas faced by doctors and researchers, and even less is known about the complex interaction between the parties within the doctor–patient relationship and its dynamic changes during the time leading up to a diagnosis. The main aim of this work is to share our experience on this issue, focusing on the perspectives of the main actors in the diagnostic process, based on the previously published literature and more than 20 years of real-life evidence in outpatient clinics and research focused on the diagnosis and management of RDs by the authors.

## 2. The “Rocky Road” of Rare Diseases

Notwithstanding all the efforts, the diagnostic success rate in RDs is still relatively low, around 40% [19]. Many reasons may underlie these figures. More than 7500 different RDs have been described so far, each affecting relatively few people [20,21,22], thus it is unlikely for many physicians to have experience with recognizing and easily managing a specific RD because they may have never encountered other patients with the same condition in their professional life. In these cases, the first diagnosis is often delayed or wrong, and optimal clinical management is seldom achieved [20]. In other cases, undiagnosed individuals may exhibit non-specific features or may be too young to reach the full expression of the disorder that will be recognized later during growth. Finally, a subject presenting with a previously unreported rare condition may be affected by multiple genetic disorders and present atypical symptoms or clinical manifestations resulting from a blended phenotype making it difficult to recognize each individual disorder [23]. On the other hand, there are still numerous technical limitations related to the genomic investigations that can be performed, and a lack of knowledge of the genetic causes underlying many RDs. Just to give a remarkable example, although rare variants in nearly 1500 genes have been shown to cause rare congenital developmental disorders, many other disease-causing genes still remain to be discovered [24,25]. When we consider the social and economic impacts, in some countries, minorities and people from different socio-demographic backgrounds experience more unequal access to opportunities to participate in clinical research programs, making it more difficult for them to receive a proper diagnosis [26]. This is in stark contrast to the principle of equity, which requires that all individuals should benefit from scientific progress, regardless of their social, economic, racial, religious, or gender background [27]. Failure to fully engage in genomic research perpetuates already significant health inequalities in these populations due to economic disparities and low accessibility to facilitators [28]. The reduced presence of diverse categories in genomics research eventually becomes a public health issue, as it diminishes the broad applicability of findings and limits the classification of rare variants in individuals from under-represented groups, making a global public health response directed towards increasing resources and access to RD research all over the world urgently needed [29,30,31]. Subjects with RDs and their families invariably experience a long medical journey from the onset of the first symptoms to the final diagnosis, which has been properly defined as a “diagnostic odyssey” by Black et al. [32]. This often implies serial referrals to a plethora of specialists and the execution of different, extensive, and possibly invasive tests before reaching the right diagnosis. It has been estimated that the average diagnostic delay is approximately 9 years with a very wide range (5 to 30 years) [23,33]. Diagnostic delay can be objectively long and perceived as intolerable, and, for some, a diagnosis may remain frustratingly elusive [3].

## 3. Too Much Time: The Reactions

Waiting time is consuming at all levels. It depletes economic resources inappropriately used in diagnostic tests or for long-term rehabilitation. This is true for both families, who, in most cases, have to afford the additional financial costs to assure their relatives a tailored rehabilitation, and for the healthcare system that has to establish a long-term follow-up and management plan without knowing the exact cause of the disorder. It drains the emotional resources compelling to rebalance patients’ and families’ lives into the dimension of a chronic illness. Eventually, it exhausts social resources. Families and affected individuals quickly realize that after a diagnosis of an RD, their world will be subverted, and the reactions of each of the players cannot be predictable, as exemplarily described by Tolstoy‘s Anna Karenina, “*all happy families are alike; each unhappy family is unhappy in its own way*” [34]. As time goes by, the involved parties implement a series of compensatory mechanisms.

RD individuals, given their unmet demands, turn to other physicians (medical shopping) with added costs on their finances and a massive emotional burden at the moment they face a growing number of disappointments when confronted with a new “*non-diagnosis*”. Considering the economic costs, the need to travel to other health centers is one of the most frequent causes of spending compounded by a delay in diagnosis [35]. In a recent review, persons who experienced diagnostic delays visited more physicians than others who were diagnosed within one year from the first symptoms, with a quarter of them visiting as many as ten specialists or more during their search for a diagnosis [35]. Mistrust of the engaged diagnostic pathway often leads to unnecessary tests being repeated, with an additional burden on healthcare costs. In a recent survey focused on Italian undiagnosed pediatric patients, the total average cost of diagnostic procedures for each undiagnosed patient was calculated to be at least EUR 11,572 with an estimated average cost for each year of diagnostic delay of EUR 2,146 [23]. Other studies also quantify an average of 7.3 medical visits and 14 tests before obtaining a diagnosis [36,37].

When the diagnosis is not achieved in a reasonable timeframe, another risk is noncompliance with symptomatic rehabilitation programs, which often represent the only helpful intervention in RDs, as they have very few curative treatment options. Noncompliance interlocks with another coping mechanism based on the false assumption that a non-diagnosis represents the absence of the disease: “*there is no diagnosis, so my child is not sick, and we do not need to rehabilitate*”. Such an adaptive strategy negatively magnifies the resulting harm to the affected subject. Denial may even be manifested in words. Families prefer not to use the terms RD and syndrome for the reason that they really want to protect their beloved from stigmatization and rejection reactions. As reported in a survey on psychological mechanisms in the reconstruction of normality after a diagnosis of a chronic childhood illness, “*When you say: he has a RD, people look at you like he has something terrible! Maybe it’s the word rare to frighten, it alerts people*” [38].

Sometimes, affected individuals and their caregivers may respond with another coping mechanism in an obsessive search for a “*culprit*” to the situation they are experiencing. Parents of a child with an RD may even recognize themselves or their partner as “*the guilty one*” (the blame), triggering disruptive mechanisms and family disaggregation [39,40,41]. A strong impact on other numerous psychosocial aspects, such as the disruption of daily routine is common in all RDs; however, diagnostic delay further impacts such dynamics, affecting at least one-third of cases [35]. In general, the period between the onset of symptoms and the diagnosis has been described as the hardest and most psychologically painful. Families perceive a disruption in normality and spend much of their time brooding on negative thoughts, such as the normality they were used to before the illness, comparing their current lives with those of others and struggling to accept the change and the loss of “*what was before*” [38]. Delays in diagnosis increase feelings of insecurity, while frequent medical check-ups and hospitalizations, as well as long journeys to specialized centers, make normal family life impossible. The longer and more complicated the diagnostic process becomes, the more stressful this period is perceived to be [38]. The diagnosis then represents a turning point that allows families to start reconstructing their real life [38]. The diagnostic delay may also explain some other additional social and emotional issues within families, at least impacting half of the families who have suffered a diagnostic delay, according to the IRDiRC definition, compared to 17.5% of those who received a diagnosis in less than one year [35]. Profound changes in the psychological health of the parents of a child with an RD have been evidenced, particularly among mothers, that eventually end up negatively influencing their relationship with the partner [42,43]. The disruption of the relationship with the partner has been claimed in at least 15% of the families with an RD and, in a minority, principally caused by a diagnostic delay [35], worsening both social and economic consequences on the familial nucleus. The one who “*pays the price*” is the affected subject since it is predictable how a disjointed core is even less able to cope with a chronic illness.

What happens to the referring physicians? Faced with a non-diagnosis, clinicians may have two approaches. They may give up by denying themselves, minimizing or underestimating symptoms with serious consequences for the patient, and indirectly hindering the definition of the natural history of the disease, and, in extreme cases, facilitating medical shopping. The other way is to accept the patient’s anxieties and fears by encouraging repeat testing, over-treatment, or alternative therapies. It has been estimated that patients with a diagnostic delay are more prone to use drugs and other treatments in the stage before diagnostic confirmation more frequently than those without a delay [35].

## 4. An Additional Fault: Uncertainty

RDs are characterized by an additional fault: uncertainty, which implies further psychological and social sequelae. It makes the disease course even more strenuous to bear and often acts as a confounding element to establishing proper rehabilitation and, in general, developing compensatory strategies for the families and the surrounding social nucleus. The sense of isolation and stigmatization that goes along with the diagnosis of an RD usually generates profound and lasting effects on the affected subjects and their caregivers. Since their recognition, which sometimes corresponds with the birth of a child, RDs are riddled with uncertainty: “*What’s wrong with me and my child? Why did it happen? How can it be solved? What can I do to feel better? What’s next? Will it happen again?*” [44]. All these demands are likely not to be unanswered for a long period after the first clinical suspicion of an RD and may generate a set of feelings (denial, disappointment, frustration, and anger) that overlap with the well-known psychological mechanisms of the process of mourning. Preventing affected individuals and families from having a proper diagnosis puts them off from moving to the next stages of this process, leaving them in the denial stage with the false expectation that “*me/my child can be saved*” and that “*I did not do enough*”.

Parents and relatives of a subject affected with an RD experience an additional psychological burden regarding their reproductive choices, as the absence of a diagnosis prevents families from making informed decisions [3].

Uncertainty also exacerbates this negative cycle for referring doctors. They find themselves in the same situation as their patients, forcing them to translate their role into a new dimension (“*I can’t cure you, but I can take care of you*”) that hardly fits the high standards of efficiency and performance to which modern medicine has accustomed us. “*I am not a good doctor*” and “*I should have diagnosed*” may represent a narcissistic wound for clinicians dealing with RDs, shattering the sense of omnipotence that prevails in the medical profession and downsizing the physician to a more human position where the care acquires more importance than the cure, the former being much more emotionally costly than the latter. Referring physicians are forced to experience (and often share) the same emotional states of patients, knowing that they will have to endure and then deliver a long series of bad news, leading to emotional overload and a risk of burnout for which they are inadequately prepared [45]. No one likes to be powerless and “*let things go*”. This process could be compared to being forced to go back in time centuries, such as having to deal with the Spanish flu without penicillin or, more recently, feeling overwhelmed and hopeless during the first wave of the COVID-19 pandemic [22,46].

Clinicians experience all the stages of mourning before their patients and often simultaneously live in frustration and anger dealing with the worsening of the disease or family recurrence of the condition. They may progressively lose contact with their patients and reframe the physician–patient relationship on the non-manifested/unspoken that allows both parties to face frustration, as there is no need to reiterate a request that cannot be answered. Mutual trust appears to be compromised, follow-up disperses over time, and parties withdraw simply because silence helps to deal with their respective failures. Finally, uncertainty may never last even after the fatal outcome of the affected individual, leaving the families in an endless limbo that worsens the grief of an unexplained loss. Death can be truly considered a watershed among families and their physicians. Families may react with closure, declining to meet again with the physician who cared for them, with the intention not to rip the old wound. Sometimes their only desire is to “*leave everything behind*” and forget what happened as an unpleasant period with the immediate effect of an initial relief, which will inevitably result in unsolved anger against themselves and the referring physician who was not able to “*save them*”.

## 5. Expectations and Hopes

When “*all seems lost*”, they then turn to research, a kind of “*last shore*” when they think that possible diagnostic or therapeutic strategies are over. Science, and the researchers who accept the request for help, are asked to do the impossible: to formulate a genetic diagnosis where routine diagnostic skills have failed, to understand the molecular and functional basis of the RD, and possibly, to suggest targeted therapeutic approaches. In other words, the researchers welcome the request to suspend the state of uncertainty and resolve the patients’ and the physicians’ queries in a hopefully “short” time, given the painful journey they have experienced before approaching an experimental research program. Sometimes turning to research might represent the only chance to access a diagnostic sequencing tool for patients from disadvantaged economic and socio-demographic backgrounds or minorities that face insurance and other barriers to receiving genomic sequencing in diagnostic clinical care [26,47].

RD patients are then turned into research patients; far from picturing themselves as frozen vials locked up in a crowded laboratory refrigerator awaiting to be processed, they should be first informed on the challenges they are about to encounter, and the energies and efforts the research team will use to reach a finding. While genuinely willing to help solve a diagnostic dilemma, researchers are challenged to combine their problems with those of the patients (and sometimes the referring physician), who are exhausted and worried about the disease they are facing, and who are often unaware (or inadequately informed) about the difficulties underlying the research effort. Researchers also experience waiting: the experimental one, aimed at identifying the genetic cause (e.g., omics sciences), followed by other waiting time filled by ad hoc built functional validation experiments, and finally, the time needed to find other individuals with a similar condition to unambiguously confirm the result and associate the candidate disease gene with the specific disorder (matchmaking). Nearly half of the cases approaching research still remain undiagnosed [4,48], and this means more waiting time during which science is looking for other approaches. Two major strategies are usually applied: the first one is the use of other OMICS analyses aimed at exploring alternative working hypotheses or solving the technical limitations of the previously used approaches [49]. However, the second action is based on reiterative data analysis that might require years of waiting time [50,51]. Time passes for researchers, too. Unlike patients (the stakeholders), and referring clinicians, who have an empathic relationship with them, researchers may be more tempted to dismiss a line of research. RDs and research focused on RDs have been correctly defined as “*orphans*” [52]. Historically, there has been limited funding for RD research for the diagnosis, treatment, and development of drugs (orphan drugs). The most common obstacles in RD research are very little/no drug companies’ interest (is it worthwhile to develop a new drug for few individuals to benefit?), technical difficulties in developing an effective drug for RDs, and final costs of the healthcare system for the drug reimbursement [53]. To this, the precariousness of research funding should be considered. Many researchers are funded by fixed-time research projects, and few dedicated investments in research are institutionalized; this ends up favoring the dismissal of a specific RD research project when time is too long and costs are not affordable anymore. Researchers are not required to “*take care of*” but are expected to fill the void left by the healthcare system in the case of an unsolved RD. Science is then left alone to cope with hope and time, and researchers are not used to it, as they do not deal with patients and do not know them.

## 6. The Tower of Babylon: Different Languages, Different Needs

In the delicate balance between the patient’s bedside and the laboratory bench, “*everyone is right*”. The affected subject and the clinician have the right to have a diagnosis (the right to ask), and the researcher has the right to act with scientific and rigorous awareness to obtain an unambiguous and reliable answer and eventually give it back to routine diagnostics (the right and the need to answer). The different languages and needs of the affected subjects, their doctors, and the researchers sometimes make it difficult, or even impossible, to dialogue, although moving toward the same goal (a diagnosis), threatening useless stresses between the sides or compromising mutual trust/compliance with damage for all. The affected subjects lose their chance to receive a diagnosis and answer to their questions, the physicians lose the opportunity to take care of their patients in the only way possible (“*I name your disease, and monitor its evolution*”), and the researchers, even when they achieve in identifying a new disease gene, risk losing the opportunity to get back to the patient and the physician and trace the natural history of the condition to be able to robustly translate their efforts into daily clinical practice. Without a shared language, we all lose and, in particular, those who “come later” lose, as an undiagnosed or inaccurately characterized RD slows the diagnosis of other individuals with the same condition and forces them into the same painful waiting time (a new diagnostic odyssey somewhere in the word).

## 7. How to Ethically Cope with Time

What can we do to overcome the failure of the therapeutic alliance and make this long waiting time more acceptable? We need someone who takes care of the waiting time. They should understand the needs of the parties and know how to negotiate, be open-minded, and have the skills to properly communicate with the parties, take into account their psychological issues, have the technical skills to integrate clinical data and genetic results but, above all, should be able to keep long and complicated disease histories straight (cold cases) by encouraging periodic re-evaluation either in research (re-analysis of omics data, joint meeting with research group and data upgrades, updates on new lines of research) and in the clinical setting (periodic follow-up with the referring clinicians, reconstruction of the natural history, prenatal genetic counseling even in the absence of a molecular diagnosis). A professional figure that might be in charge of this role is already present in many healthcare services all over the world and is represented by the genetic counselor. They are usually healthcare professionals that work alongside medical geneticists to provide information to patients, prepare them for genetic and genomic testing, help them during the decisional process, and support patients and their families over the longer term [54]. Specific boards and societies officially represent them [55,56,57,58]. These precious and supportive professionals are called to answer the demands for information and support aspects during and after the diagnostic process. Some countries have officially included this professional in clinical genetics services; unfortunately, there are many other countries where this role has not yet been recognized. Genetic counselors should be included in all the genetic services managing patients with RDs. Alongside the genetic counselors, we need a “*linking role*” who can understand and alternatively speak with the patient, the clinician, and the research team, who translates the results for each of them and explains the waiting time, making it more acceptable. They should be also able to facilitate the creation of psychological support groups for parents and families of subjects with an RD and for the other professionals involved in the diagnostic process. We are aware of and recommend the institution of this coordinating position with a high professional and human profile that can give (and also have the strength not to give) the best answers to all the parties involved. Finally, geneticists and researchers should be financially supported through the establishment of dedicated multidisciplinary research networks focused on RD and undiagnosed patients and based on the use of genome sequencing, which should be strongly considered in the planning of national and international health policy strategies.

Many diagnostic therapeutic healthcare pathways (PDAs) exist for the most frequent diseases. RDs, while rare in their singularity, are very frequent in the aggregate, affecting 6–8% of the general population, thus representing a public health problem [29,59]. A dedicated shared PDA should be generated to guarantee equal and free access to research for all RD patients. A healthcare system that requires competence, efficiency, and high-quality standards should consider caring for individuals with an RD whose perspectives are unlikely to be a curative therapy and whose prognosis is often worsening and sometimes fatal in an alternative way. We should interact with them and facilitate their journey toward knowledge of and coping with the disease, creating a simple space with a simple and effective language.

## 8. Conclusions

The authors of the manuscript come from the ends of this space where the affected subjects, their diseases, histories, and life experiences are placed, and they daily find themselves dealing with the waiting time. As in Alice in Wonderland, nothing is forever, and time is relative, mutually changing, and extremely long or short. Individuals with RDs, their physicians, and researchers well fit into the Alice and White Rabbit paradox: “*How long is forever?—Alice said. White Rabbit—sometimes, just one second*” [60].

## Data Availability

No new data were generated.

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
