# Peer review of "The Right to Ask, the Need to Answer—When Patients Meet Research: How to Cope with Time"

_ijerph, 2023, doi:10.3390/ijerph20054573_

Round 1

Reviewer 1 Report

1.     Line 21 - edit to either “in each stakeholder “or “in each stakeholder group.” Also delete “with” in next sentence.

2.     I don’t agree with the authors’ concluding sentence of the abstract. “best” is relative and may be different for different stakeholders. For example, while waiting, families might find advocacy gratifying rather than simply waiting.

3.     Line 43 – on what do they base their assessment that rare disease are “almost always” genetic? Perhaps a better word would be “frequently.”

4.     Line 46 -delete “on”

5.     Line 65 – encountered

6.     Line 87-88 – “This is true for both families, who in most cases have to afford the additional financial…” A family is a “who not a “that,” and I think they mean “afford” related to costs, not “effort.” 

7.     Line 90 – insert “plan” after “management” and replace “to” with “the” emotional…

8.     Lina 91 is not clear in its’ meaning. Do they mean resources instead of sources in this and line 92?

9.     Line 94 exemplarily is the correct form of the word…

10.  Line 98 – deal is not the correct word…maybe burden, or weight?

11.  Line 111 the higher risk…higher than what? This is a non-qualified statement and needs a reference.

12.  There appear to be many unsupported opinions in the paper, especially paragraph starting with line 123. 

13.  Line 152-153 “facilitating dismiss” – not clear what this means

14.  Lina 235 – what is meant by “more prone to dismiss a research line?” to stop working on a project? Edit paragraph to be more clear.

15.  Line 178 “pour” – not sure what this means.

16.  Lines 220- 230 – lots of grammar errors and lack of clarity

17.  Paragraph on ethically coping – long discussion of what is needed, but no comment on the existence of professionals who already fill this role in many institutions – genetic counselors and study coordinators.

18.  Article needs general and extensive editing for correct English grammar and word usage. 

19.  Article is striking for the absence of comment on diversity and the relative access to the benefits of technologies by non-white and other underserved/minority populations.

20.  No comment on the benefit of support organizations, even general organizations,  that are so helpful to families

21. There are no real conclusions or resolutions presented. I understand the this is primarily an opinion piece, but comments that will add to the literature, such as creative solutions, are needed.

Author Response

We wish to thank the reviewer for her constructive remarks.

 Remarks:

1. Line 21 - edit to either “in each stakeholder “or “in each stakeholder group.” Also delete “with” in next sentence

3. Line 43 – on what do they base their assessment that rare disease are “almost always” genetic? Perhaps a better word would be “frequently.” 

4. Line 46 -delete “on”

5. Line 65 – encountered

6. Line 87-88 – “This is true for both families, who in most cases have to afford the additional financial…” A family is a “who not a “that,” and I think they mean “afford” related to costs, not “effort.” 

7. Line 90 – insert “plan” after “management” and replace “to” with “the” emotional…

8. Line 91 is not clear in its’ meaning. Do they mean resources instead of sources in this and line 92?

9. Line 94 exemplarily is the correct form of the word…

10. Line 98 – deal is not the correct word…maybe burden, or weight?

11. Line 111 the higher risk…higher than what? This is a non-qualified statement and needs a reference. 

13. Line 152-153 “facilitating dismiss” – not clear what this means

14. Line 235 – what is meant by “more prone to dismiss a research line?” to stop working on a project? Edit paragraph to be more clear

15. Line 178 “pour” – not sure what this means

16. Lines 220- 230 – lots of grammar errors and lack of clarity

18. Article needs general and extensive editing for correct English grammar and word usage

Authors’ reply: We thank the reviewer for her accurate editing of our original version of the manuscript. We took into account all points. Following the reviewer’s advice, we revised the text for correct English grammar and word usage.

2. I don’t agree with the authors’ concluding sentence of the abstract. “best” is relative and may be different for different stakeholders. For example, while waiting, families might find advocacy gratifying rather than simply waiting

Authors’ reply: We thank the reviewer for the comment. Family associations and patients support organizations undoubtedly represent a powerful and important resource. However, the aim of this contribution is to suggest a complementary support to them by physicians and researchers to help to manage the waiting time. Our work was intended to explore the perspectives of the main actors during  the diagnostic process and suggest a way to ameliorate the relationship among the parties involved in diagnosis. We mainly focused on how much time influences compliance in the diagnostic pathway. If the parties do not know mutual needs, they may mistrust and break the therapeutic alliance. Following the reviewer’s remark, we revised the sentence to be clear that physicians and researchers are in charge to cope with time in order to care for patients.

12. There appear to be many unsupported opinions in the paper, especially paragraph starting with line 123

Authors’ reply: As previously clarified, our work was mainly focused on the “real life” experience of the authors in order to contribute to the previous literature on exploring coping reactions in families and caregivers after a diagnosis in RD. To clear this point, we added a paragraph in the “background” section and stated it. We also slightly rephrased the sentence at line 123. We would like to underline that these “opinions” are not exclusively based on personal experience. Consistently, we added literature (refs 38 to 43) for the reactions from parents, including the one referring to sentence in line 123, as well as physicians’ reactions and coping mechanisms (including higher risk of burn out) (see refs 44 to 46). In the revised version, the paragraph is not unsupported by literature anymore.

  1. Paragraph on ethically coping – long discussion of what is needed, but no comment on the existence of professionals who already fill this role in many institutions – genetic counselors and study coordinators.

Authors’ reply: We thank the reviewer for her precious comment. Indeed, the genetic counselor is a professional figure that may help in dealing waiting time. However, although this healthcare professional is institutionalized in some countries, this figure is not present in many other realities, including many European countries and the developing ones. We commented on it in the paragraph and added references about discrepancies.

  1. Article is striking for the absence of comment on diversity and the relative access to the benefits of technologies by non-white and other underserved/minority populations.

Authors’ reply: We thank the reviewer for the valuable comment. In our first version, we decided to focus on the difficulties after entering in research programs rather than the difficulty to enter in these dedicated programs. Following the reviewer’s advice, we added a paragraph in the “rocky road” section and stressed that research might be considered an opportunity for minorities to be diagnosed in the “expectation and hopes” section.

  1. No comment on the benefit of support organizations, even general organizations,  that are so helpful to families.

Authors’ reply: The reviewer’s remark is logical, too. However, this is an article about interaction among different realities (patients, caregivers, physicians and researchers), their complex reactions, and consequences on failure of therapeutic alliance. While important in principle, the role of PAOs is beyond the topic of this manuscript.

  1. There are no real conclusions or resolutions presented. I understand this is primarily an opinion piece, but comments that will add to the literature, such as creative solutions, are needed.

Reviewers’ reply: As the reviewer properly commented, our work is an opinion piece. However, we added some solutions, such as the creation of a psychological support group for all the involved parties, the need of an institutionalized government funding for rapid diagnostic research, and the creation of a shared diagnostic therapeutic care pathways PDA for RDs.

Reviewer 2 Report

The article under the title: "The right to ask, the need to answer. When patients meet research: how to cope with time" raises the very important psychological-ethical problem of diagnosis and any related problems of rare diseases which are associated with handicaps which require dedicated therapy. The authors have shown all the problems that lead to frustration, especially for parents, but also for scientists, biologists and doctors.

What I missed in this article was a broad perspective on how exactly it should work in such a situation to best help parents of children with rare genetic diseases and how to intensify research so that even in such cases we can at least identify the genetic cause of the disease and further try to find an individual pharmacotherapy based on genetic knowledge.

Such points in my opinion include:

1) The creation of national and international centres for rare genetic diseases with multi-institute teams for the comprehensive treatment of rare diseases.

2) Creation of an internationally accessible gene bank with a rapid exchange of information.

3) government funding of rapid diagnostic research groups, with particular emphasis on next generation sequencing.

4) Creation of psychological support groups for parents and families of children with rare genetic diseases.

5) Discussions with the pharmaceutical industry on the need to work on drugs, genetic therapies for rare diseases. Discussions should also include the cost of possible downstream therapies, e.g. gene therapy. It may be financially justifiable and due to the enormous effort currently put into the treatment of children with spinal muscular atrophy (SMA), where the cost of therapy is estimated at 2 million dollars, but it is unethical to offer such an expensive therapy, where parents often have to look for funding sources. The state has a responsibility to help such parents. After the trauma of the long diagnostic path and the associated psychological problems of the parents of such a sick child, it is not possible to offer therapy with a chance of a cure that is not affordable.

I consider this article to be very valuable and important, even if it concerns a few percent of the society. All the more so it should be discussed and written about, because it is necessary to create regulations and protocols with the international impact.

Author Response

We wish to thank the reviewer for the positive feedback and constructive input.

 Remarks:

  1. The creation of national and international centres for rare genetic diseases with multi-institute teams for the comprehensive treatment of rare diseases

 Authors’ reply: We thank the reviewer for this comment, which was extensively reviewed in the “background” section with corresponding literature about educational programs and conjoined projects to increase diagnostic rate and improve the diagnostic path.

  1. Creation of an internationally accessible gene bank with a rapid exchange of information.

Authors’ reply: In our first version, we commented on this point (please also see refs 3, 4, 21, 22, 49-51 of the revised manuscript) and explained it in the background.

  1. Government funding of rapid diagnostic research groups, with particular emphasis on next generation sequencing.

Authors’ reply: This would represent a really helpful solution. We thank the reviewer for this suggestion. We added it into the “conclusion” section.

  1. Creation of psychological support groups for parents and families of children with rare genetic diseases

Authors’ reply: Again, we thank the reviewer for this valuable suggestion. We added it into the “conclusion” section as well.

  1. Discussions with the pharmaceutical industry on the need to work on drugs, genetic therapies for rare diseases. Discussions should also include the cost of possible downstream therapies, e.g. gene therapy. It may be financially justifiable and due to the enormous effort currently put into the treatment of children with spinal muscular atrophy (SMA), where the cost of therapy is estimated at 2 million dollars, but it is unethical to offer such an expensive therapy, where parents often have to look for funding sources. The state has a responsibility to help such parents. After the trauma of the long diagnostic path and the associated psychological problems of the parents of such a sick child, it is not possible to offer therapy with a chance of a cure that is not affordable.

Authors’ reply: This is a big ELSI issue that, however, is beyond the aims of this work. We completely agree with the reviewer, and we cited this problem into the “research” section of our original version of the manuscript. However, we think that this work might not be suitable to extensively comment on this problem. A dedicated article should be considered.

Round 2

Reviewer 1 Report

I appreciate the additions you have made to the paper, especially the information about underserved communities and available professional roles. While I recognize that this is in large part an opinion piece, I continue to be concerned about how you mis-represent the many roles of genetic counselors. I am a genetic counselor  working in research, and have colleagues who do what I do, and I work to do all the things you describe and attribute to a highly professional and "human" individual. I agree with your premise, but I think these individuals already exist, perhaps not in all countries but in many, and training programs are expanding to provide adequate coverage for the expanding number of positions needing such an individual.

The document still needs some editing for English and I have noted some but not all of these below. There are also some sentences that require clarification as to your meaning.

Comments on Version 2

1.     Editing for correct English still needed; I’ve noted only a few  in this list.

2.     Line 48-49 " and increasing access to genetic diagnostics through extensive patients and physicians empowerment" – not sure what this means

3.     Paragraph starting line 98 – thanks for adding this!

4.     Line 120 – sources or resources?

5.     Line 166: disruption in normality

6.     Line 203 – do you want to say : “doomed to be unanswered” or relate the learning curve about natural history of disease to the delay in diagnosis and diagnostic odyssey. The goal is to answer these questions, not to doom them to un-answerability. 

7.     Line 217 – wondering if current medical training still leads MDs to feel omnipotent. If so, it’s not being realistic…

8.     Line 259: researchers are challenged to confront their problems with those of the patients (and sometimes the referring physician),,,, Is confront the correct word here? Do the authors mean "combine" or something else? The statement is not clear.

9.     Line 271 researchers more inclined or forced by financial circumstances to dismiss

10.  Line 321 – genetic counselors are non-MD but not non-medical. Genetic counselors do have specific medical training to allow them to facilitate many aspects of genetic medical care.

11.  Line 327: I understand that this is an opinion piece, but the statement “they may be not adequately prepared to cope with frustration 327 and to mediate among the parties during the long waiting time” does not acknowledge the psychocsocial and advocacy training that genetic counselors received. I think they are uniquely positioned to serve this role.

12.  Lines 334-338> many genetic counselors serve in this role in academia and in industry.

Author Response

We wish to thank the reviewer for her constructive remarks.

General comment. “I appreciate the additions you have made to the paper, especially the information about underserved communities and available professional roles. While I recognize that this is in large part an opinion piece, I continue to be concerned about how you mis-represent the many roles of genetic counselors. I am a genetic counselor working in research, and have colleagues who do what I do, and I work to do all the things you describe and attribute to a highly professional and "human" individual. I agree with your premise, but I think these individuals already exist, perhaps not in all countries but in many, and training programs are expanding to provide adequate coverage for the expanding number of positions needing such an individual.”

Authors’ reply: We are grateful to the reviewer for her careful assessment of our manuscript. From the discussion, we realize that managing this relevant and complex topic is variably approached by the different National Health Systems. Overall, we agree with the reviewer regarding the key and relevant role of the genetic counselor, are really hope this figure will be included in all countries in the next years. Currently, the situation is quite heterogeneous. We revised the manuscript to emphasize these considerations.

1. “Editing for correct English still needed; I’ve noted only a few in this list

2. “Line 48-49 "and increasing access to genetic diagnostics through extensive patients and physicians empowerment" – not sure what this means”

4. “Line 120 – sources or resources?

5. “Line 166: disruption in normality.”

8. “Line 259: researchers are challenged to confront their problems with those of the patients (and sometimes the referring physician). Is confront the correct word here? Do the authors mean "combine" or something else? The statement is not clear.”

9. “Line 271 researchers more inclined or forced by financial circumstances to dismiss”

10. “Line 321 – genetic counselors are non-MD but not non-medical. Genetic counselors do have specific medical training to allow them to facilitate many aspects of genetic medical care.”

Authors’ reply: We thank the reviewer for her accurate editing of our revised version of the manuscript. We took into account all points. Following the reviewer’s advice, we revised the text for correct English grammar and word usage.

6. “Line 203 – do you want to say: “doomed to be unanswered” or relate the learning curve about natural history of disease to the delay in diagnosis and diagnostic odyssey. The goal is to answer these questions, not to doom them to un-answerability.”

Authors’ reply: We agree with the reviewer that the final goal in RDs management is to answer these questions. Unfortunately, as the reviewer herself is conscious, being a genetic counselor, parents and affected individuals highly suffer from feelings that resemble a mourning when dealing with a clinical suspicion of RD. The diagnostic delay to reach a definite diagnosis is highly probable to worsen these feelings. To clear this point we revised the sentence as follows: “All these demands are likely not to be unanswered for a long period after the first clinical suspicion of a RD”.

7. “Line 217 – wondering if current medical training still leads MDs to feel omnipotent. If so, it’s not being realistic…”

Authors’ reply: We politely disagree with the reviewer. A somehow “sense of omnipotence” has been always characterized the medical profession since its birth. The current technological progresses and potentials have widely increased trust on the medical profession and on its “saving role” to heal patients. This is true for most of diseases, so that general population and physicians, as well, are accustomed that nowadays almost every patient can “be saved” somehow. A wide literature is present in Pubmed on Medical students’ feelings (please see for a comprehensive review the BMC educational work by Tempski et al. 2012, What do medical students think about their quality of life? A qualitative study. https://doi.org/10.1186/1472-6920-12-106).  Physicians are no exception to this rule, especially in subspecialities, such as surgery, transplants or infertility treatment (see Fedele et al. 2020, What About Fertility Staff Emotions? An Explorative Analysis of Healthcare Professionals’ Subjective Perspective, PMID: 33680202). End-life care, emergency and resuscitation departments, oncology, and genetics are true exceptions and we are forced to face frustration and failure every day. No dedicated educational program to face this huge burden is established during the medical training and we have to learn how to cope with it after we graduated. Maybe the reviewer is accustomed to work with MDs trained by experience to it.

11. “Line 327: I understand that this is an opinion piece, but the statement “they may be not adequately prepared to cope with frustration 327 and to mediate among the parties during the long waiting time” does not acknowledge the psychosocial and advocacy training that genetic counselors received. I think they are uniquely positioned to serve this role.”

Authors’ reply: We agree with the reviewer. Genetic counselors are indeed positioned to serve this role. As we might guess, the reviewer comes from a reality in which the genetic counselor profession is a well established one, and there is an adequate coverage for the expanding number of positions needing them. This is not true for all countries. In our piece, our intention was to point to those countries in which this professional figure is not established at all or is not properly formed/trained. Following the reviewer’s advice, we removed the sentence and reinforced the concept of the need of such a figure in healthcare genetic services.

12. “Lines 334-338. Many genetic counselors serve in this role in academia and in industry.” 

Authors’ reply: again, we agree with the reviewer, but she should keep in mind that the whole word is not a single country. Many realities must be taken into account (as the reviewer properly asked us to consider minorities and disadvantaged populations) when we talk about international health policy strategies.
